# DIAR: Diffusion-model-guided Implicit Q-learning with Adaptive Revaluation

## Abstract

We propose a novel offline reinforcement learning (offline RL) approach, introducing the Diffusion-model-guided Implicit Q-learning with Adaptive Revaluation (DIAR) framework. We address two key challenges in offline RL: out-of-distribution samples and long-horizon problems. We leverage diffusion models to learn state-action sequence distributions and incorporate value functions for more balanced and adaptive decision-making. DIAR introduces an Adaptive Revaluation mechanism that dynamically adjusts decision lengths by comparing current and future state values, enabling flexible long-term decision-making. Furthermore, we address Q-value overestimation by combining Q-network learning with a value function guided by a diffusion model. The diffusion model generates diverse latent trajectories, enhancing policy robustness and generalization. As demonstrated in tasks like Maze2D, AntMaze, and Kitchen, DIAR consistently outperforms state-of-the-art algorithms in long-horizon, sparse-reward environments.

## 1 Introduction

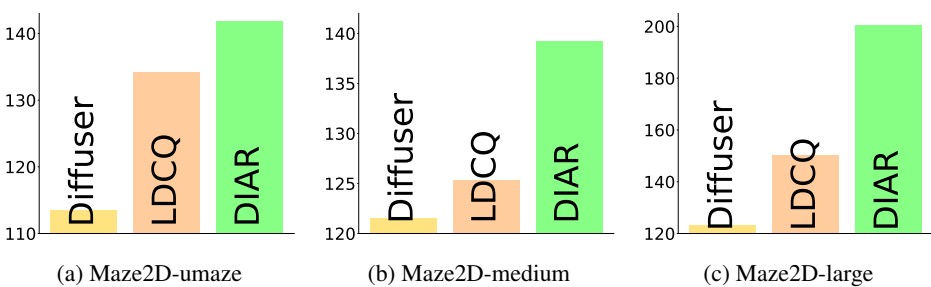

(a) Maze2D-umaze         (b) Maze2D-medium         (c) Maze2D-large

Figure 1: Performance comparison across D4RL environments with long-horizon and sparse-reward tasks, specifically Maze2D. Our method, DIAR, consistently outperforms other diffusion-based planning frameworks, including Diffuser and LDCQ.

Offline reinforcement learning (offline RL) is a type of reinforcement learning where an agent learns a policy using pre-collected datasets instead of gathering data through direct interactions with the environment (Fujimoto et al., 2019). By avoiding real-world interactions, offline RL eliminates safety concerns. Furthermore, it makes efficient use of the collected data, which is especially beneficial when gathering new data is costly or time-consuming. However, offline RL depends on the dataset, meaning the policy it learns may perform poorly if the data is low quality or biased. Moreover, a distributional shift can occur during the process of learning from offline data (Levine et al., 2020), leading to degraded performance in the real environment.

To overcome the limitations of offline RL, existing research have been made to address these issues by leveraging diffusion models, a type of generative model (Janner et al., 2022). Incorporating diffusion models allows for learning the overall distribution of the state and action spaces, allowing decisions to be made based on this knowledge. Methods such as Diffuser (Janner et al., 2022) and Decision Diffuser (DD) (Ajay et al., 2023) use diffusion models to predict decisions not autoregressively one step at a time, but instead by inferring the entire decision for the length of the

horizon at once, achieving strong performance in long-horizon tasks. Additionally, methods like LDCQ (Venkatraman et al., 2024) propose using latent diffusion models to learn the Q-function, allowing the Q-function to make more appropriate predictions for out-of-distribution state-actions.

Recent studies of diffusion-based offline RL methods, often bypass the use of the Q-function or rely on other offline Q-learning methods (Janner et al., 2022; Ajay et al., 2023). However, recent research has proposed a novel approach that does not avoid the Q-function but instead leverages diffusion models to assist in Q-learning (Wang et al., 2023; Venkatraman et al., 2024). This approach enables handling a wide range of Q-values for diverse states and actions. We found that using samples generated by diffusion models can improve the agent's performance.

Therefore, we propose Diffusion-model-guided Implicit Q-learning with Adaptive Revaluation (DIAR), which integrates the value function and data sampled from the diffusion model into the training and decision process. This approach provides more objective assessment of the current state, enabling the Q-function to achieve a balance between long-horizon decision-making and step-by-step refinement. In the training process, the Q-function and value function alternates between learning from the dataset and samples generated by the diffusion model, allowing it to adapt to a wide variety of scenarios. Additionally, value function also helps to reevaluate the current decision to explore new action sequences and select a more optimal path.

DIAR consistently outperforms existing offline RL algorithms, especially in environments that involve complex route planning and long-horizon state-action pairs like Figure 2. Additionally, as shown in Figure 1, DIAR achieves state-of-the-art performance in environments such as Maze2D, AntMaze, and Kitchen (Fu et al., 2020). This research highlights the potential of diffusion models to enhance both policy abstraction and adaptability in offline RL, with significant implications for real-world applications in robotics and autonomous systems.

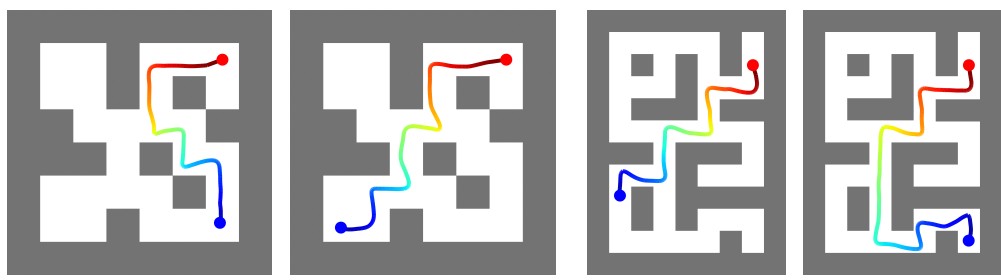

(a) Maze2D-medium hard cases        (b) Maze2D-large hard cases

Figure 2: DIAR-generated trajectories in challenging Maze2D situations. DIAR reliably reaches the goal even from starting points (blue) that are far from the goal (red). DIAR shows strong performance regardless of starting position.

## 2 RELATED WORK

### 2.1 OFFLINE REINFORCEMENT LEARNING

Offline reinforcement learning (offline RL), also referred to as batch reinforcement learning, has gained significant attention in recent years due to its potential in learning effective policies from pre-collected datasets without further interaction with the environment. This paradigm is particularly useful in real-world applications where exploration can be costly or dangerous, such as healthcare, robotics (Kalashnikov et al., 2018), and autonomous driving.

One of the primary challenges in offline RL is the issue of out-of-distribution actions (Kumar et al., 2019), where a learned policy selects actions not well represented in the offline dataset. To address this, several works have introduced behavior regularization techniques that constrain the policy to remain close to the behavior policy seen in the offline data. Among these, Conservative Q-learning (CQL) introduces a conservative Q-function that underestimates the value of out-of-distribution actions, reducing the likelihood of the learned policy selecting potentially harmful actions (Kumar et al., 2020). By minimizing the overestimation of value functions, CQL facilitates more reliable

policy learning from offline data. Another notable approach is Implicit Q-learning (IQL), which implicitly regularizes the learned Q-function by keeping it close to the empirical value of the actions observed in the dataset (Kostrikov et al., 2022). This prevents the over-optimization of Q-values for actions that are rarely or never observed in the offline dataset. Additionally, Batch-Constrained Q-learning (BCQ) imposes direct limitations on the learned policy to prevent deviations from the actions observed in the offline dataset (Fujimoto et al., 2019). BCQ introduces a constraint that ensures the learned policy selects actions similar to the behavior policy, thus avoiding the exploitation of inaccurate Q-value estimates for unseen actions.

## 2.2 DIFFUSION-BASED PLANNING IN OFFLINE RL

Diffusion models (Sohl-Dickstein et al., 2015; Ho et al., 2020) have shown remarkable performance in fields such as image inpainting (Lugmayr et al., 2022) and image generation (Ramesh et al., 2022; Saharia et al., 2022). Recent research has extended the application of diffusion models beyond image domains to address classical trajectory optimization challenges in offline RL. One prominent model, Diffuser (Janner et al., 2022), directly learns trajectory distributions and generates tailored trajectories based on situational demands. By prioritizing trajectory accuracy over single-step precision, Diffuser mitigates compounding errors and adapts to novel tasks or goals unseen during training. Additionally, Decision Diffuser (DD) was introduced, which predicts the next state using a state diffusion model and leverages inverse dynamics for decision-making (Ajay et al., 2023). Furthermore, a method called Latent Diffusion-Constrained Q-learning (LDCQ) has been proposed, which combines latent diffusion models with Q-learning to reduce extrapolation errors (Venkatraman et al., 2024). Emerging methods also focus on learning interpretable skills from visual and language inputs and applying conditional planning via diffusion models (Liang et al., 2024). Approaches that generate goal-divergent trajectories using Gaussian noise and facilitate reverse training through denoising processes have also been explored (Jain & Ravanbakhsh, 2023).

## 3 PRELIMINARY: LATENT DIFFUSION REINFORCEMENT LEARNING

To train the Q-network, a diffusion model that has trained based on latent representations is required. The first step is to learn how to represent an action-state sequence of length $H$ as a latent vector using $\beta$-Variational Autoencoder ($\beta$-VAE) (Pertsch et al., 2021). The second step is to train the diffusion model using the latent vectors generated by the encoder of the $\beta$-VAE. This allows the diffusion model to learn the latent space corresponding to the action-state sequence. Subsequently, the Q-network is trained using the latent vectors generated by the diffusion model.

**Latent representation by $\beta$-VAE**    The $\beta$-VAE plays three key roles in the initial stage of our model training. First, the encoder $q_{\theta_E}(\boldsymbol{z}|\boldsymbol{s}_{t:t+H}, \boldsymbol{a}_{t:t+H})$ must effectively represent the action-state sequence $\boldsymbol{s}_{t:t+H}, \boldsymbol{a}_{t:t+H}$ from the dataset $\mathcal{D}$ into a latent vector $\boldsymbol{z}$. Second, the distribution of $\boldsymbol{z}$ generated by the $\beta$-VAE must be conditioned by the state prior $p_{\theta_s}(\boldsymbol{z}|\boldsymbol{s}_t)$. This is learned by minimizing the KL-divergence between the latent vector generated by the encoder and the one generated by the state prior. The formation of the latent vector is controlled by adjusting the $\beta$ value, which determines the weight of KL-divergence. Lastly, the policy decoder $\pi_{\theta_D}(\boldsymbol{a}_t|\boldsymbol{s}_t, \boldsymbol{z})$ of the $\beta$-VAE must be able to accurately decode actions when given the current state and latent vector as inputs. These three objectives are combined to train the $\beta$-VAE by maximizing the evidence lower bound (ELBO) (Kingma & Welling, 2014) as shown in Eq. 1.

$$\mathcal{L}(\theta) = \mathbb{E}_{\mathcal{D}}\big[\mathbb{E}_{q_{\theta_E}}\big[\sum_{i=t}^{t+H-1} \log \pi_{\theta_D}(\boldsymbol{a}_i|\boldsymbol{s}_i, \boldsymbol{z})\big] - \beta D_{KL}(q_{\theta_E}(\boldsymbol{z}|\boldsymbol{s}_{t:t+H}, \boldsymbol{a}_{t:t+H}) \parallel p_{\theta_s}(\boldsymbol{z}|\boldsymbol{s}_t))\big] \quad (1)$$

**Training latent vector with a diffusion model**    The latent diffusion model (LDM) effectively learns latent representations, focusing on the latent space instead of the original data samples (Rombach et al., 2022). The model minimizes a loss function that predict the initial latent $\boldsymbol{z}_t$ generated by the VAE encoder $q_\phi$, rather than noise as in traditional diffusion models. $H$-length trajectory segments $\boldsymbol{s}_{t:t+H}, \boldsymbol{a}_{t:t+H}$ are sampled from dataset $\mathcal{D}$ and paired with initial states and latent variables $(\boldsymbol{s}_t, \boldsymbol{z}_t)$. The focus lies on modeling the prior $p(\boldsymbol{z}|\boldsymbol{s}_t)$ to capture the distribution of latent $\boldsymbol{z}$

given the state $s_t$. A conditional latent diffusion model $\mu_\psi(z|s_t)$ is utilized and refined with a time-dependent denoising function $\mu_\psi(z^j, s_t, j)$ to reconstruct $z^0$ through the denoising step $j \sim [1, T]$. Consequently, the LDM is trained by minimizing the loss function $\mathcal{L}(\psi)$ as given in Eq. 2.

$$\mathcal{L}(\psi) = \mathbb{E}_{j \sim [1,T], (s,a) \sim \mathcal{D}, z_t \sim q_\phi(z|s,a), z^j \sim \mu_\psi(z^j|z^0)} \left( \| z_t - \mu_\psi(z^j, s_t, j) \|^2 \right) \tag{2}$$

**Q-network by latent representation**    To train the Q-network, Eq. 3 reduces extrapolation errors by restricting policy updates to the empirical distribution of the offline dataset (Venkatraman et al., 2024). Prioritizing trajectory accuracy over single-step precision allows the model to mitigate compounding errors and remain adaptable to novel tasks or goals unseen during training. Furthermore, the integration of temporal abstraction and latent space modeling notably enhances the mechanisms underlying credit assignment and improves the effectiveness of policy optimization.

$$Q(s_t, z_t) \leftarrow Q(s_t, z_t) + \alpha \left[ r_{t:t+H} + \gamma \max_{z'_{t+H} \sim \mu_\psi} Q(s_{t+H}, z'_{t+H}) - Q(s_t, z_t) \right] \tag{3}$$

The latent vector $z'_{t+H}$ generated by the diffusion model is utilized in the training of the Q-function. The Q-function learns the relation between the $Q(s_{t+H}, z'_{t+H})$ and $Q(s_t, z_t)$ like Eq. 3, which are based on the initial state $s_t$ and latent vector $z_t$ pairs present in the dataset, and the $z'_{t+H}$ generated by the diffusion model. $r_{t:t+H}$ denotes the sum of rewards with discount factor $\gamma$. This enables the model to adapt to new tasks or goals that were not observed in the offline data. Furthermore, the integration of temporal abstraction and latent space modeling significantly enhances the mechanism of credit assignment, thereby improving the effectiveness of policy optimization. According to Eq. 3, the trained Q-function is used such that, as shown in Eq. 4, when a state $s_t$ is given, the decision is made by selecting the action that has the highest Q-value.

$$\pi(s_t) = \pi_\theta(a_t | \arg\max_{z_i \sim \mu_\psi(z|s_t)} Q(s_t, z_i)) \tag{4}$$

## 4    PROPOSED METHOD

Using diffusion models to address long-horizon tasks typically involves training over the full trajectory length (Janner et al., 2022). This approach differs from autoregressive methods that focus on selecting the best action at each step, as it learns the entire action sequence over the horizon. This allows the model to learn long sequences of decisions at once and generate a large number of actions in a single pass. However, predicting decisions across the entire horizon may not always lead to optimal outcomes, as the primary goal is to generate a sequence of decisions corresponding to the sequence length.

Additionally, there is a well-known problem of overestimating the Q-value when training a Q-network (Hasselt et al., 2016; 2018; Fu et al., 2019; Kumar et al., 2019; Agarwal et al., 2020). This occurs when certain actions, appearing intermittently, are assigned a high $Q(s, a)$ value. In these cases, the state may not actually hold high value, but the Q-value becomes "lucky" and inflated. Therefore, it is essential to ensure that the Q-network does not overestimate and can correctly assess the value based on the current state.

To resolve both of these issues, we propose Diffusion-model-guided Implicit Q-learning with Adaptive Revaluation (DIAR), introducing a value function to assess the value of each situation. Unlike the Q-network, which learns the value of both state and action, the state-value function learns only the value of the state. By introducing constraints from the value function, we can train a more balanced Q-network and, during the decision-making phase, make more optimal predictions with the help of the value function.

### 4.1    DIFFUSION-MODEL-GUIDED Q-LEARNING FRAMEWORK

The value-network $V_\eta$ with parameter $\eta$ evaluates the value of the current state $s_t$, and the Q-network $Q_\phi$ with parameter $\phi$ evaluates the value of the current state $s_t$ and action $a_t$. Additionally, by

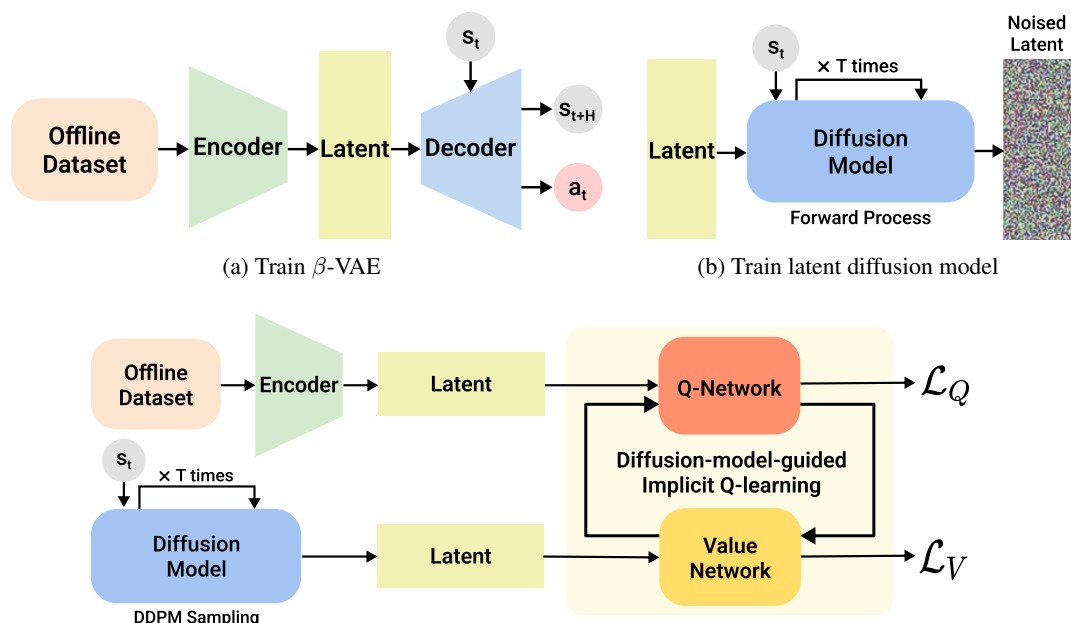

(a) Train $\beta$-VAE

(b) Train latent diffusion model

(c) Train Q-network and value-network

Figure 3: Three training stages of DIAR. (a) The $\beta$-VAE is trained by encoding a state-action sequence spanning an $H$-length horizon into a latent space, followed by a policy decoder that outputs actions based on the encoded latent $z$ and the state $s_t$ contained within it. (b) A diffusion model is trained using the encoded latent and the initial state $s_t$. (c) The Q-network is trained on the offline dataset, while the value network is trained on data generated by the diffusion model. This interplay allows the value function and Q-function to guide each other, enabling more balanced learning across both offline samples and generated data.

combining value-network learning with Q-network learning, constraints can be applied to the Q-network, resulting in more balanced training. Instead of relying on the dataset to train the value network and Q-network, we enhance the process by introducing latent vectors generated through a diffusion model. By doing so, we minimize extrapolation errors for unseen decisions in the dataset, leading to more accurate value estimation.

The training of the value-network should aim to reduce the difference between the Q-value and the state-value. Therefore, it is crucial to include the difference between $Q(s, z)$ and $V(s)$ in the loss function. To achieve this, rather than simply using MSE loss, we apply weights to make the data distribution more flexible and to respond more sensitively to differences. We use an asymmetric weighted loss function that multiplies the weights of variables $u$ by an expectile factor $\tau$, as shown in Eq. 5. In the next step, $u$ is used as the difference between the Q-value and the state-value for loss calculation.

$$L_\tau^2(u) = |\tau - \mathbb{I}(u < 0)|u^2 \tag{5}$$

By using an asymmetrically weighted loss function, the value-network is trained to reduce the difference between the Q-value and the state-value. We set $\tau$ to a value greater than 0.5 and apply Eq. 6 to assign more weight when the difference between the Q-value and the value is large. Additionally, instead of using latent vector encoded from the dataset, we use latent vectors $\tilde{z}_t$ generated by the diffusion model to guide the learning of a more generalized Q-network.

$$L_V(\eta) = \mathbb{E}_{s_t \sim \mathcal{D}, \, \tilde{z}_t \sim \mathcal{D}_\psi} \left[ L_\tau^2 \left( Q_{\hat{\phi}}(s_t, \tilde{z}_t) - V_\eta(s_t) \right) \right] \tag{6}$$

After the loss for the value-network is calculated, the loss for the Q-network is computed. The loss in Eq. 7 is not based on the Q-network alone but is learned based on the current value and

reward, ensuring balance with the value network. The value-network learning, using latent vectors generated by the diffusion model, allows it to handle diverse trajectories, while the Q-network is trained on data pairs $(s_t, z_t, r_{t:t+H}, s_{t+H}) \sim \mathcal{D}$ from the dataset, learning the Q-value of state-latent vector pairs based on existing trajectories. Q-network and value-network training processes form a complementary relationship.

$$L_Q(\phi) = \mathbb{E}_{(s_t, z_t, r_{t:t+H}, s_{t+H}) \sim \mathcal{D}} \left[ (r_{t:t+H} + \gamma V_\eta(s_{t+H}) - Q_\phi(s_t, z_t))^2 \right] \tag{7}$$

To ensure stable Deep Q-network training and prevent Q-value overestimation, we employed the Clipped Double Q-learning method (Fujimoto et al., 2018). Additionally, we used a prioritized replay buffer $\mathcal{B}$, where the Q-network is trained based on the priority of the samples (Schaul et al., 2016). $\mathcal{B}$ stores $(s_t, z_t, r_{t:t+H}, s_{t+H})$, which are generated from the offline dataset. The state, action, and reward are taken from the offline dataset, and the latent vector $z_t$ is encoded by $q_{\theta_E}(z|s, a)$. The encoded latent vector $z_t$, along with the current state $s_t$, is used to guide the MLP model through the diffusion model to learn the Q-value. The Q-network and value-network are trained alternately, maintaining a complementary relationship through their respective loss functions. The value-network's loss $L_V(\eta)$ is calculated based on the difference between the Q-value and the state-value, which is adjusted by the expectile factor $\tau$. The Q-network's loss $L_Q(\phi)$ is computed using the Bellman equation with the reward and value, where the effect of distant timesteps is controlled by the discount factor $\gamma$. The calculated Q-network loss $L_Q(\phi)$ is updated in the model $Q_\phi$ via backpropagation, and the target Q-network $Q_{\hat{\phi}}$ is gradually updated based on the update rate $\rho$. The detailed process can be found in Algorithm 1.

---

**Algorithm 1:** Diffusion-model-guided Implicit Q-learning with Adaptive Revaluation

1 **Input:** Q-network $Q_\phi$, target Q-network $Q_{\hat{\phi}}$, value-network $V_\eta$, diffusion model $\mu_\psi(z|s)$, prioritized replay buffer $\mathcal{B}$, horizon $H$, number of sampling latent vectors $n$, latent vector $z$, update rate $\rho$, max iteration $T$, learning rate $\lambda_Q, \lambda_V$

2 $\hat{\phi} \leftarrow \phi$
3 $t \leftarrow 0$
4 **while** $t < T$ **do**
5     $(s_t, z_t, r_{t:t+H}, s_{t+H}) \leftarrow \mathcal{B}$
6     $z_{t+H}^0, z_{t+H}^1, \ldots, z_{t+H}^{n-1} \leftarrow \mu_\psi(z|s_{t+H})$             # Sampling $n$ latent vectors
7     $\eta \leftarrow \eta - \lambda_V \nabla_\eta L_V(\eta)$                     # Training value-network
8     $\phi \leftarrow \phi - \lambda_Q \nabla_\phi L_Q(\phi)$                    # Training Q-network
9     $\hat{\phi} \leftarrow \rho\phi + (1-\rho)\hat{\phi}$
10     Update priority of $\mathcal{B}$
11 **end**

---

## 4.2 ADAPTIVE REVALUATION IN POLICY EXECUTION

DIAR method reforms a decision if the value of the current state is higher than the value of the state after making a decision over the horizon length $H$. We refer to this process as Adaptive Revaluation. Using the value-network $V_\eta$, if the current state's value $V(s_t)$ is greater than $V(s_{t+H})$, the value after making a decision for $H$ steps, the method generates a new latent vector $z_t$ from the current state $s_t$ and continues the decision-making process. When predicting over the horizon length, there may be cases where taking a different action midway through the horizon would be more optimal. In such cases, the value-network $V_\eta$ checks this, and if the condition is met, a new latent vector is generated.

Adaptive Revaluation uses the difference in value to examine whether the agent's predicted decision is optimal. Since the current state $s_t$ can be obtained directly from the environment, it is easy to compute the value $V(s_t)$ of the current state $s_t$. Whether the current trajectory is optimal can be determined using a state decoder $f_\theta(s_{t+H}|s_t, z_t)$. By inputting the current state $s_t$ and latent vector

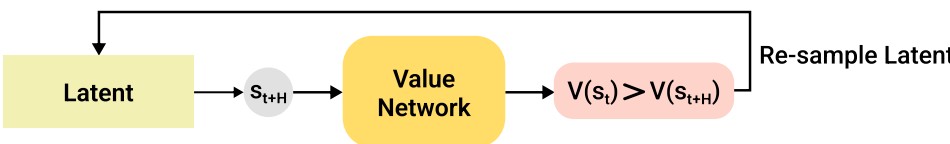

Figure 4: Inference step with DIAR. The current state $s_t$ is put into the diffusion model to extract candidate latent vectors. Then, the latent vector $z_t$ with the highest $Q(s_t, z_t)$ is selected as the best latent vector. This latent vector $z_t$ is subsequently decoded to generate the action $a_t$. Additionally, the future state $s_{t+H}$ is also decoded to be used for calculating the future value $V(s_{t+H})$.

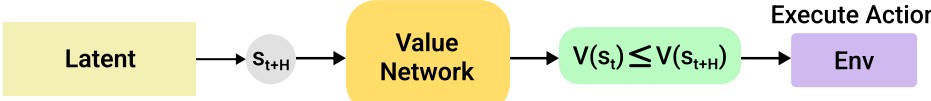

(a) Non-ideal values for states within a latent in the sparse-reward environment

(b) Ideal values for states within a latent in the sparse-reward environment

Figure 5: The process of finding a better trajectory using Adaptive Revaluation. The process involves making a decision and taking action based on the skill latent $z_t$ with the highest $Q(s_t, z_t)$. The current latent vector $z_t$ is used to predict the future state $s_{t+H}$, based on which the value $V(s_{t+H})$ of the future state $s_{t+H}$ is calculated. (a) If the value $V(s_t)$ of the current state $s_t$ is greater than the value $V(s_{t+H})$ of the future state $s_{t+H}$, it is considered non-ideal, and re-sampling is performed. (b) If the value $V(s_{t+H})$ of the future state $s_{t+H}$ is greater than or equal to the value $V(s_t)$ of the current state $s_t$, it is considered ideal, and the action $a_t$ decoded by the latent vector $z_t$ is executed continuously.

$z_t$ into the state decoder, the future state $s_{t+H}$ can be predicted. This predicted $s_{t+H}$ is passed into the value-network $V_\eta$ to estimate its future value $V(s_{t+H})$. By comparing these two values, if the current value $V(s_t)$ is higher, the agent generates new latent vectors and selects the one with the highest $Q(s_t, z_t)$. The detailed Adaptive Revaluation algorithm is shown in Appendix B.

## 4.3 THEORETICAL ANALYSIS OF DIAR

In this section, we prove that in the case of sparse rewards, when the current timestep $t$, if the value $V(s_t)$ of the current state $s_t$ is higher than the value $V(s_{t+H})$ of the future state $s_{t+H}$, there is a more ideal trajectory than the current trajectory. An ideal trajectory is defined as one where, for all states at timestep $k$, the discount factor $0 < \gamma \le 1$ ensures that $V(s_k) \le V(s_{k+1})$. This means that for an agent performing actions toward a goal, the value of each state in the trajectory increases monotonically.

Now, consider an assumption about an ideal trajectory: for any timesteps $i, j$ with $i < j$, we assume that $V(s_i) > V(s_j)$ for $s_i$ and $s_j$ from the dataset $\mathcal{D}$. Furthermore, since the state $s_j$ is not the goal and we are in a sparse reward setting, $\forall r(s_i, a_i) = 0$. If we write the Bellman equation for the value function, it results in Eq. 8.

$$V(\boldsymbol{s}_i) = \mathbb{E}_{(\boldsymbol{s}_i, \boldsymbol{a}_i, \boldsymbol{s}_{i+1}) \sim \mathcal{D}} \left[ r(\boldsymbol{s}_i, \boldsymbol{a}) + \gamma V(\boldsymbol{s}_{i+1}) \right] \tag{8}$$

Eq. 8 represents the value function $V(\boldsymbol{s}_i)$ when there is a difference of one timestep. The value function $V(\boldsymbol{s}_i)$ can be computed using the reward received from the action taken in the current state $\boldsymbol{s}_i$ and the value of the next state $\boldsymbol{s}_{i+1}$. Therefore, by iterating Eq. 8 to express the timesteps from $i$ to $j$, we obtain Eq. 9.

$$V(\boldsymbol{s}_i) = \mathbb{E}_{(\boldsymbol{s}_{i:j}, \boldsymbol{a}_{i:j}) \sim \mathcal{D}} \left[ \sum_{t=i}^{j-1} \gamma^{t-i} r(\boldsymbol{s}_t, \boldsymbol{a}_t) + \gamma^{j-i} V(\boldsymbol{s}_j) \right] \tag{9}$$

Since the current environment is sparse in rewards, no reward is given if the goal is not reached. Therefore, in Eq. 9, all reward $r(\boldsymbol{s}_t, \boldsymbol{a}_t)$ terms are zero. By substituting the reward as zero and reorganizing Eq. 9, we can derive Eq. 10.

$$V(\boldsymbol{s}_i) = \mathbb{E}_{(\boldsymbol{s}_{i:j}, \boldsymbol{a}_{i:j}) \sim \mathcal{D}} \left[ \gamma^{j-i} V(\boldsymbol{s}_j) \right] \tag{10}$$

Since the magnitude of $\gamma$ is $0 < \gamma \leq 1$, the term $\gamma^{j-i} V(\boldsymbol{s}_j)$ is always less than or equal to $V(\boldsymbol{s}_j)$. This contradicts the initial assumption, indicating that the assumption is incorrect. Therefore, for any ideal trajectory, all value functions $V(\boldsymbol{s}_i)$ must follow a monotonically increasing function. In other words, if the trajectory predicted by the agent is an ideal trajectory, the value $V(\boldsymbol{s}_j)$ after making a decision over the horizon $H$ must always be greater than the current value $V(\boldsymbol{s}_i)$. If the current value $V(\boldsymbol{s}_i)$ is greater than the future value $V(\boldsymbol{s}_j)$, then this trajectory is not an ideal trajectory. Consequently, generating a new latent vector $\boldsymbol{z}_i$ from the current state $\boldsymbol{s}_i$ to search for an optimal decision is a better approach.

## 5 EXPERIMENTS

We compare the performance of our model with other models under various conditions and environments. We focus on goal-based tasks in environments with long-horizons and sparse rewards. For offline RL, we use the Maze2D, AntMaze, and Kitchen datasets to test the strengths of our model in long-horizon sparse reward settings (Fu et al., 2020). These environments feature very long trajectories in their datasets, and rewards are only given upon reaching the goal, making them highly suitable for evaluating our model. We also compare the performance improvements achieved when using Adaptive Revaluation, analyzing whether it allows for reconsideration of decisions when incorrect ones are made and enables the generation of the correct trajectory. Furthermore, to ensure more accurate performance measurements, all scores are averaged over 100 runs and repeated 5 times, with the mean and standard deviation reported.

### 5.1 PERFORMANCE ON OFFLINE RL BENCHMARKS

In this section, we compare the performance of our model in offline RL. To evaluate our model, we compare it against various state-of-the-art models. These include behavior cloning (BC), which imitates the dataset, and offline RL methods based on Q-learning, such as IQL (Kostrikov et al., 2022) and IDQL (Hansen-Estruch et al., 2023). We also compare our model with DT (Chen et al., 2021), which uses the transformer architecture employed in LLMs, and methods that use diffusion models, such as Diffuser (Janner et al., 2022), DD (Ajay et al., 2023), and LDCQ (Venkatraman et al., 2024). Through these comparisons with various algorithms, we conduct a quantitative performance evaluation of our model.

Datasets like Maze2D and AntMaze require the agent to learn how to navigate from a random starting point to a random location. Simply mimicking the dataset is insufficient for achieving good performance. The agent must learn what constitutes a good decision and how to make the best judgments throughout the trajectory. Additionally, the ability to stitch together multiple paths through trajectory combinations is essential. In particular, the AntMaze dataset involves a complex state space and requires learning and understanding high-dimensional policies. We observed that our method DIAR, consistently demonstrated strong performance in these challenging tasks, where

Table 1: Comparison with other methods in long horizon sparse reward D4RL environments.

| Dataset | BC | IQL | DT | IDQL | Diffuser | DD | LDCQ | DIAR |
|---|---|---|---|---|---|---|---|---|
| maze2d-umaze-v1 | 3.8 | 47.4 | 27.3 | 57.9 | 113.5 | - | 134.2 | 141.8±4.3 |
| maze2d-medium-v1 | 30.3 | 34.9 | 32.1 | 89.5 | 121.5 | - | 125.3 | 139.2±3.5 |
| maze2d-large-v1 | 5.0 | 58.6 | 18.1 | 90.1 | 123.0 | - | 150.1 | 200.3±3.4 |
| antmaze-umaze-diverse-v2 | 45.6 | 62.2 | 54.0 | 62.0 | - | - | 81.4 | 88.8±1.5 |
| antmaze-medium-diverse-v2 | 0.0 | 70.0 | 0.0 | 83.5 | 45.5 | 24.6 | 68.9 | 68.2±6.7 |
| antmaze-large-diverse-v2 | 0.0 | 47.5 | 0.0 | 56.4 | 22.0 | 7.5 | 57.7 | 60.6±2.4 |
| kitchen-complete-v0 | 65.0 | 62.5 | - | - | - | - | 62.5 | 68.8±2.1 |
| kitchen-partial-v0 | 38.0 | 46.3 | 42.0 | - | - | 57.0 | 67.8 | 63.3±0.9 |
| kitchen-mixed-v0 | 51.5 | 51.0 | 50.7 | - | - | 65.0 | 62.3 | 60.8±1.4 |

high-dimensional abstraction and reasoning are critical. For more demonstrations, please refer to the Appendix F.

## 5.2 IMPACT OF ADAPTIVE REVALUATION

In this section, we analyze the impact of Adaptive Revaluation. We directly compare the cases where Adaptive Revaluation is used and not used in our model. The test is conducted on long-horizon sparse reward tasks, where rewards are sparse. For overall training, an expectile value of $\tau = 0.9$ was used, with $H = 30$ for Maze2D and $H = 20$ for AntMaze and Kitchen. Other training settings were generally the same, and detailed configurations can be found in the Appendix A.

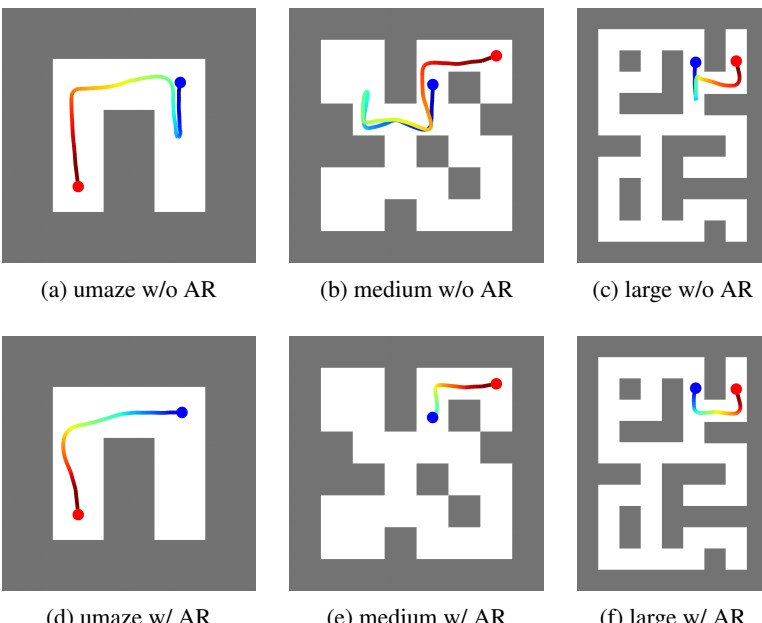

| (a) umaze w/o AR | (b) medium w/o AR | (c) large w/o AR |
|---|---|---|

| (d) umaze w/ AR | (e) medium w/ AR | (f) large w/ AR |
|---|---|---|

Figure 6: (a)∼(c) Three Maze2D results that only the Q-function is used without Adaptive Revaluation. (d)∼(f) Three Maze2D results for improved decision making using Adaptive Revaluation. Even without Adaptive Revaluation, our model performs well, but we can observe that using Adaptive Revaluation enables more efficient decision-making.

When Adaptive Revaluation is used, it checks whether a better decision might exist according to the value function and discovers a better latent vector to re-create the trajectory. If the value of the current state is higher than the value of a future state, it indicates that a better trajectory might exist than the currently selected decision. This enables the agent to choose a more accurate abstraction and form a more optimal trajectory based on it. The improvement in decision-making with Adaptive Revaluation can be observed in Table 2, which shows how much the agent's decisions improve when using this method.

Table 2: Comparison of performance changes with Adaptive Revaluation (AR) in D4RL tasks.

| Dataset | DIAR w/o AR | DIAR w/ AR |
|---|---|---|
| maze2d-umaze-v1 | 135.6±2.8 | 141.8±4.3 |
| maze2d-medium-v1 | 138.2±3.1 | 139.2±3.5 |
| maze2d-large-v1 | 193.5±4.7 | 200.3±3.4 |
| antmaze-umaze-diverse-v2 | 88.8±1.5 | 85.4±2.6 |
| antmaze-medium-diverse-v2 | 68.2±6.7 | 67.4±3.4 |
| antmaze-large-diverse-v2 | 56.0±4.6 | 60.6±2.4 |
| kitchen-complete-v0 | 68.8±2.1 | 63.8±3.0 |
| kitchen-partial-v0 | 63.3±0.9 | 63.0±2.5 |
| kitchen-mixed-v0 | 60.0±0.7 | 60.8±1.4 |

## 5.3 COMPARISON WITH SKILL LATENT MODELS

We further compare our model with other reinforcement learning methods that use skill latents. For the D4RL tasks, we selected methods that use generative models to learn skills and make decisions based on them. As performance baselines, we chose the VAE-based methods OPAL[1] (Ajay et al., 2021) and PLAS (Zhou et al., 2020), as well as Flow2Control (Yang et al., 2023), which utilizes normalizing flows. The performance comparison is shown in Table 3.

Table 3: Performance comparison with other skill latent learning methods in D4RL tasks.

| Dataset | BC | PLAS | IQL+OPAL | Flow2Control | DIAR |
|---|---|---|---|---|---|
| maze2d-umaze-v1 | 3.8 | 57.0 | - | - | 141.8±4.3 |
| maze2d-medium-v1 | 30.3 | 36.5 | - | - | 139.2±3.5 |
| maze2d-large-v1 | 5.0 | 122.7 | - | - | 200.3±3.4 |
| antmaze-umaze-diverse-v2 | 45.6 | 45.3 | 70.2 | 81.6 | 88.8±1.5 |
| antmaze-medium-diverse-v2 | 0.0 | 0.7 | 42.8 | 83.7 | 68.2±6.6 |
| antmaze-large-diverse-v2 | 0.0 | 0.0 | 52.4 | 52.8 | 60.6±2.4 |
| kitchen-complete-v0 | 65.0 | 34.8 | 11.5 | 75.0 | 68.8±2.1 |
| kitchen-partial-v0 | 38.0 | 43.9 | 72.5 | 74.9 | 63.3±0.9 |
| kitchen-mixed-v0 | 51.5 | 40.8 | 65.7 | 69.2 | 60.8±1.4 |

## 6 CONCLUSION

In this study, we proposed Diffusion-model-guided Implicit Q-learning with Adaptive Revaluation (DIAR), which leverages diffusion models to improve abstraction capabilities and train more adaptive agents in offline RL. First, we introduced an Adaptive Revaluation algorithm based on the value function, which allows for long-horizon predictions while enabling the agent to flexibly revise its decisions to discover more optimal ones. Second, we propose an Diffusion-model-guided Implicit Q-learning. Offline RL faces the limitation of difficulty in evaluating out-of-distribution state-action pairs, as it learns from a fixed dataset. By leveraging the diffusion model, a generative model, we balance the learning of the value function and Q-function to cover a broader range of cases. By combining these two methods, we achieved state-of-the-art performance in long-horizon sparse reward tasks such as Maze2D, AntMaze, and Kitchen. Our approach is particularly strong in long-horizon sparse reward situations, where it is challenging to assess the current value. Additionally, a key advantage of our method is that it performs well without requiring extensive hyper-parameter tuning for each task. We believe that the latent diffusion model holds significant strengths in offline RL and has high potential for applications in various fields such as robotics.

---

[1]To compare its effect on implicit learning, we refer to the results from Yang et al. (2023).

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

## A  EXPERIMENTS DETAILS

DIAR consists of three main components: the $\beta$-VAE for learning latent skills, the latent diffusion model for learning distributions through latent vectors, and the Q-function, which learns the value of state-latent vector pairs and selects the best latent. These three models are trained sequentially, and when learning the same task, the earlier models can be reused. Detailed model settings and hyperparameters are discussed in the next section. For more detailed code implementation and process, you can refer directly to the code on GitHub.

### A.1  $\beta$-VARIATIONAL AUTOENCODER

The $\beta$-VAE consists of an encoder, policy decoder, state prior, and state decoder. The encoder uses two stacked bidirectional GRUs. The output of the GRU is used to compute the mean and standard deviation. Each GRU output is passed through an MLP to calculate the mean and standard deviation, which are then used to compute the latent vector. This latent vector is used by the state prior, state decoder, and policy decoder. The policy decoder takes the latent vector and the current state as input to predict the current action. The state decoder takes the latent vector and the current state to predict the future state. Lastly, the state prior learns the distribution of the latent vector for the current state, ensuring that the latent vector generated by the encoder is trained similarly through KL divergence.

In Maze2D, $H = 30$ is used; in AntMaze and Kitchen, $H = 20$ is used. The diffusion model for the diffusion prior used in $\beta$-VAE training employs a transformer architecture. This model differs from the latent diffusion model discussed in the next section, and they are trained independently. Training the $\beta$-VAE for too many epochs can lead to overfitting of the latent vector, which can negatively impact the next stage.

Table 4: Hyperparameters for VAE training

| Hyperparameter | Value |
|---|---|
| Learning rate | 5e-5 |
| Batch size | 128 |
| Epochs | 100 |
| Latent dimension | 16 |
| $\beta$ | 0.1 |
| Diffusion prior steps | 200 |
| Optimizer | Adam |

### A.2  LATENT DIFFUSION MODEL

The generative model plays the role of learning the distribution of the latent vector for the current state. The current state and latent vector are concatenated and then re-encoded for use. The architecture of the diffusion model follows a U-Net structure, where the dimensionality decreases and then increases, with each block consisting of residual blocks. Unlike the traditional approach of predicting noise $\epsilon$, the diffusion model is trained to directly predict the latent vector $z$. This process is constrained by Min-SNR-$\gamma$. Overall, the diffusion model operates similarly to the DDPM method.

Table 5: Hyperparameters for Diffusion model training

| Hyperparameter | Value |
|---|---|
| Learning rate | 1e-4 |
| Batch size | 128 |
| Epochs | 450 |
| Diffusion steps | 500 |
| Drop probability | 0.1 |
| Min-SNR ($\gamma$) | 5 |
| Optimizer | Adam |

## A.3 Q-LEARNING

In our approach, we utilize both a Q-network and a Value network. The Q-network follows the DDQN method, employing two networks that learn slowly according to the update ratio. The Value network uses a single network. Both the Q-network and the Value network are structured with repeated MLP layers. The Q-network encodes the state into a 256-dimensional vector and the latent vector into a 128-dimensional vector. These two vectors are concatenated and passed through additional MLP layers to compute the final Q-value. The Value network only encodes the state into a 256-dimensional vector, which is then used to compute the value. Between the linear layers, GELU activation functions and LayerNorm are applied. In this way, both the Q-network and Value network are implicitly trained under the guidance of the diffusion model.

Table 6: Hyperparameters for Q-learning

| Hyperparameter | Value |
|---|---|
| Learning rate | 5e-4 |
| Batch size | 128 |
| Discount factor ($\gamma$) | 0.995 |
| Target network update rate | 0.995 |
| PER buffer $\alpha$ | 0.7 |
| PER buffer $\beta$ | $0.3 \rightarrow 1$ |
| Number of latent samples | 500 |
| Expectile ($\tau$) | 0.9 |
| extra steps | 5 |
| Scheduler | StepLR |
| Scheduler step | 50 |
| Scheduler $\gamma$ | 0.3 |
| Optimizer | Adam |

# B  DIAR POLICY EXECUTION DETAILS

We provide a detailed explanation of how DIAR performs policy execution. It primarily selects the latent with the highest Q-value. However, if the current state value $V(s_{t+h})$ is higher than the future state value $V(s_{s+H})$, it triggers another search for a new latent. DIAR repeats this process until it either reaches the goal or the maximum step $T$ is reached.

---

**Algorithm 2:** DIAR Policy Execution

---

1 **Input:** environment *Env*, Q-network $Q(s, a)$, value-network $V(s)$, policy decoder
$\pi_{\theta_D}(a|s, z)$, state decoder $f_\theta(s_{t+H}|s_t, z_t)$, diffusion model $\mu_\psi(z|s)$, horizon *H*, max step *T*,
number of sampling latent vectors *n*, latent vector $z$

2 $t \leftarrow 0$
3 $done \leftarrow False$
4 **while** not $done$ **do**
5     $s_t \leftarrow Env$
6     $z_t^0, z_t^1, \ldots, z_t^{n-1} \leftarrow \mu_\psi(z|s)$          # Sampling latents vectors from diffusion model
7     $Q(s_t, z_t^0), Q(s_t, z_t^1), \ldots, Q(s_t, z_t^{n-1}) \leftarrow Q_\eta(s, z)$          # Calculate Q value
8     $z_t^i \leftarrow \underset{z_t^i}{\arg\max} \, Q(s_t, z_t^i), \; z^i \in \{z_t^0, z_t^1, \ldots z_t^{n-1}\}$
9     $s_{t+H} \leftarrow f_\theta(s_{t+H}|s_t, z_t^i)$          # Predict future state
10    $V(s_{t+H}) \leftarrow V_\phi(s)$          # Calculate value of future state
11    $h \leftarrow 0$
12    **for** $h < H$ **do**
13       $s_{t+h} \leftarrow Env$
14       $V(s_{t+h}) \leftarrow V_\phi(s)$          # Calculate value of current state
15       **if** $V(s_{t+H}) < V(s_{t+h})$ **then**
16         **break**
17       **end**
18       **else**
19         $a_{t+h} \leftarrow \pi_{\theta_D}(a_{t+h}|s_{t+h}, z_t^i)$
20         Execute action $a_{t+h}$
21         Update $done$ by *Env*
22         $h \leftarrow h + 1$
23       **end**
24    **end**
25    $t \leftarrow t + h$
26 **end**

---

# C  TRAINING PROCESS FOR $\beta$-VAE

This section details the process by which the $\beta$-VAE is trained. The $\beta$-VAE consists of four models: the skill latent encoder, policy decoder, state decoder, and state prior. These four components are trained simultaneously. Additionally, a diffusion prior is trained alongside to guide the $\beta$-VAE in generating appropriate latent vectors. The detailed process can be found in Algorithm 3.

---

**Algorithm 3:** Training Beta Variational Autoencoder

---

1  **Input:** Dataset $\mathcal{D}$, state $s_t$, action $a_t$, epoch $M$, horizon $H$, diffusion steps $T$, Min-SNR $\gamma$, state prior $p_{\theta_s}(z_t|s_t)$, latent encoder $q_{\theta_E}(z_t|s_{t:t+H}, a_{t:t+H})$, policy decoder $\pi_{\theta_D}(a_{t+i}|s_{t+i}, z_t)$, state decoder $f_\theta(s_{t+H}|s_t, z_t)$, $\beta$-VAE parameter $\theta$, diffusion prior $\mu_\psi$, KL regularization coefficient $\beta$

2  $iter \leftarrow 0$

3  **for** $iter < M$ **do**

4  $\quad s_{t:t+H}, a_{t:t+H} \leftarrow \mathcal{D}$

5  $\quad z_t \leftarrow q_{\theta_E}(z_t|s_{t:t+H}, a_{t:t+H})$          # Encoding latent vector

6  $\quad \mathcal{L}_1 \leftarrow -\sum_{i=0}^{H-1} \log \pi_{\theta_D}(a_{t+i}|s_{t+i}, z_t)$          # Reconstruction loss

7  $\quad \mathcal{L}_2 \leftarrow D_{KL}(q_{\theta_E}(z_t|s_{t:t+H}, a_{t:t+H}) \| p_{\theta_s}(z_t|s_t))$      # KL divergence with state prior

8  $\quad \mathcal{L}_3 \leftarrow -\log f_\theta(s_{t+H}|s_t, z_t)$          # State decoder loss

9  $\quad$ Noise latents $z_j$ from Gaussian noise, $j \sim \mathcal{U}[1, T]$

10 $\quad \mathcal{L}_4 \leftarrow \min\{\text{SNR}(j), \gamma\}(\|z_t - \mu_\psi(z_j, s_t, j)\|^2)$          # Diffusion prior loss

11 $\quad \mathcal{L}_{total} \leftarrow \mathcal{L}_1 + \beta\mathcal{L}_2 + \mathcal{L}_3 + \mathcal{L}_4$

12 $\quad$ Update $\theta$ to minimize $\mathcal{L}_{total}$

13 $\quad iter \leftarrow iter + 1$

14 **end**

---

# D TRAINING PROCESS FOR LATENT DIFFUSION MODEL

This section also provides an in-depth explanation of how the latent diffusion model is trained. The goal of the latent diffusion model is to learn the distribution of latent vectors generated by the $\beta$-VAE. The latent diffusion model is trained by first converting the offline dataset into latent vectors using the encoder of the $\beta$-VAE, and then learning from these latent vectors. The detailed process can be found in Algorithm 4.

---

**Algorithm 4:** Training Latent Diffusion Model

---

1 **Input:** Dataset $\mathcal{D}$, state $s_t$, action $a_t$, epoch $M$, horizon $H$, diffusion steps $T$, Min-SNR $\gamma$, latent encoder $q_{\theta_E}(\boldsymbol{z}|\boldsymbol{s}, \boldsymbol{a})$, diffusion model $\mu_\psi$, variance schedule $\alpha_1, \ldots \alpha_T, \bar{\alpha}_1, \ldots \bar{\alpha}_T, \beta_1, \ldots \beta_T$

2 $iter \leftarrow 0$

3 **for** $iter < M$ **do**

4    $\boldsymbol{s}_{t:t+H}, \boldsymbol{a}_{t:t+H} \leftarrow \mathcal{D}$

5    $\boldsymbol{z}_t \leftarrow q_{\theta_E}(\boldsymbol{z}_t|\boldsymbol{s}_{t:t+H}, \boldsymbol{a}_{t:t+H})$           # Encoding latent vector

6    Sample diffusion time $j \sim \mathcal{U}[1, T]$

7    Noise latents from Gaussian noise $\boldsymbol{z}_j \sim \mathcal{N}(\sqrt{\bar{\alpha}_j}\boldsymbol{z}_t, (1 - \bar{\alpha}_j)\mathbf{I})$

8    $\mathcal{L} \leftarrow \min\{\text{SNR}(j), \gamma\}(\|\boldsymbol{z}_t - \mu_\psi(\boldsymbol{z}_j, \boldsymbol{s}_t, j)\|^2)$        # Diffusion model loss

9    Update $\psi$ to minimize $\mathcal{L}$

10    $iter \leftarrow iter + 1$

11 **end**

---

# E  DIFFUSION PROBABILISTIC MODELS

Diffusion models (Sohl-Dickstein et al., 2015; Ho et al., 2020) function as latent variable generative models, formally expressed through the equation $p_\theta(x_0) := \int p_\theta(x_{0:T}), dx_{1:T}$. Here, $x_1, \ldots, x_T$ denote the sequence of latent variables, integral to the model's capacity to assimilate and recreate the intricate distributions characteristic of high-dimensional data types like images and audio. In these models, the forward process $q(x_t|x_{t-1})$ methodically introduces Gaussian noise into the data, adhering to a predetermined variance schedule delineated by $\beta_1, \ldots, \beta_T$. This step-by-step addition of noise outlines the approximate posterior $q(x_{1:T}|x_0)$ within a structured mathematical formulation, which is specified as follows:

$$q(x_{1:T}|x_0) := \prod_{t=1}^{T} q(x_t|x_{t-1}), \quad q(x_t|x_{t-1}) := \mathcal{N}(x_t; \sqrt{1-\beta_t}x_{t-1}, \beta_t I) \tag{11}$$

The iterative denoising process, also known as the reverse process, enables sample generation from Gaussian noised data, denoted as $p(x_T) = \mathcal{N}(x_T; 0, I)$. This process is modeled using a Markov chain, where each step involves generating the sample of the subsequent stage from the sample of the previous stage based on conditional probabilities. The joint distribution of the model, $p_\theta(x_{0:T})$, can be represented as follows:

$$p_\theta(x_{0:T}) := p(x_T) \prod_{t=1}^{T} p_\theta(x_{t-1}|x_t), \quad p_\theta(x_{t-1}|x_t) := \mathcal{N}(x_{t-1}; \mu_\theta(x_t, t), \Sigma_\theta(x_t, t)) \tag{12}$$

In the Diffusion Probabilistic Model, training is conducted via a reverse process that meticulously reconstructs the original data from noise. This methodological framework allows the Diffusion model to exhibit considerable flexibility and potent performance capabilities. Recent studies have further demonstrated that applying the diffusion process within a latent space created by an autoencoder enhances fidelity and diversity in tasks such as image inpainting and class-conditional image synthesis. This advancement underscores the effectiveness of latent space methodologies in refining the capabilities of diffusion models for complex generative tasks (Rombach et al., 2022). In light of this, the application of conditions and guidance to the latent space enable diffusion models to function effectively and to exhibit strong generalization capabilities.

# F  QUALITATIVE DEMONSTRATION THROUGH MAZE2D RESULTS

Following the main section, we report more results in the Maze2D environments. We qualitatively demonstrate that DIAR consistently generates favorable trajectories.

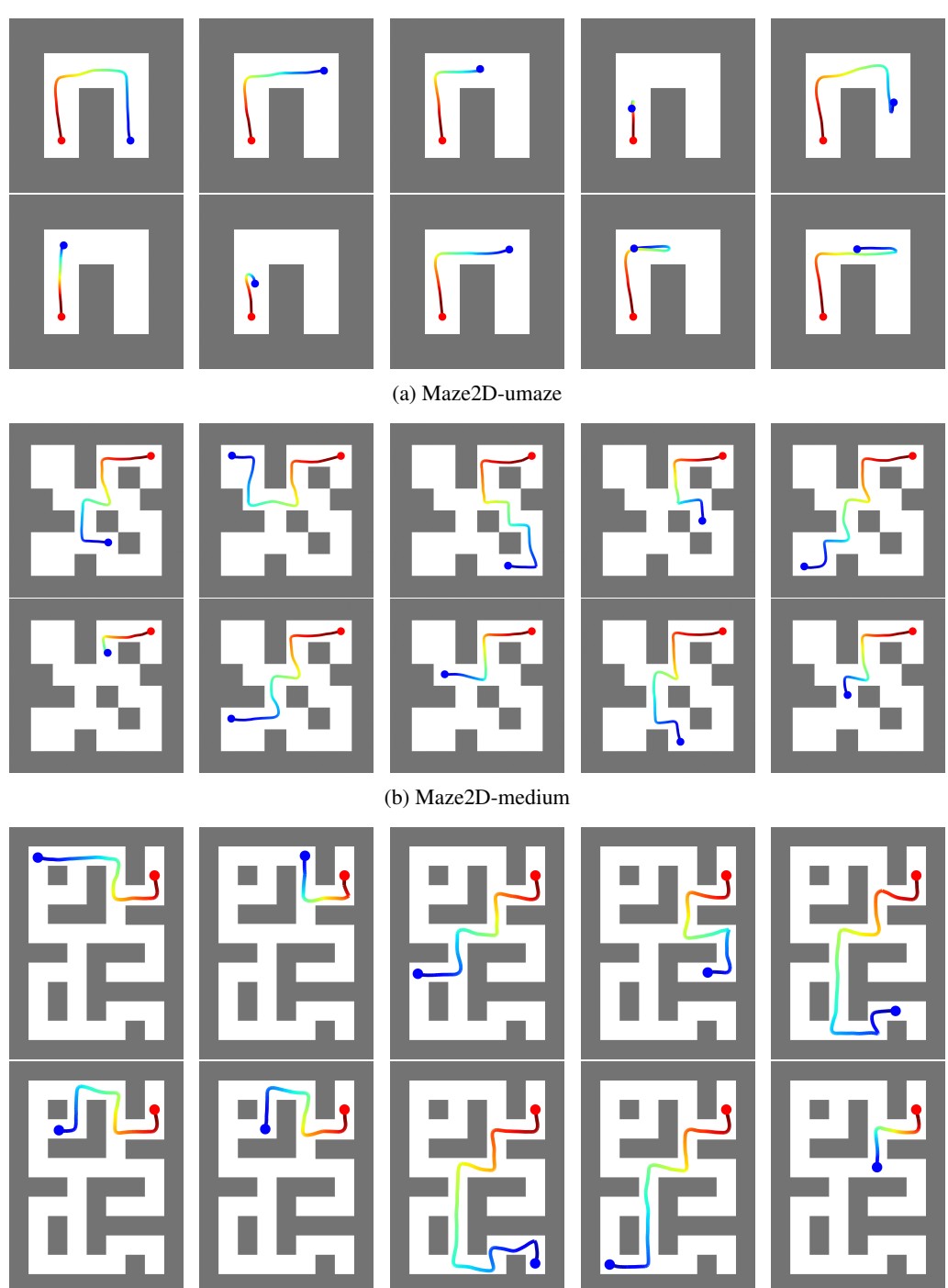

(a) Maze2D-umaze

(b) Maze2D-medium

(c) Maze2D-large

Figure 7: DIAR-generated trajectories in diverse Maze2D demonstration. DIAR reliably reaches the goal even from starting points (blue) that are far from the goal (red). It even exhibits significant advantages in cases where decisions involve longer horizons.

