# OpenReview forum: "DIAR: Diffusion-model-guided Implicit Q-learning with Adaptive Revaluation"
_ICLR.cc/2025/Conference — Submitted to ICLR 2025_

### Official Review · Reviewer_y9BV · 2024-10-19

**Soundness:** 2
**Presentation:** 3
**Contribution:** 2
**Rating:** 3
**Confidence:** 3

**Summary:**

Offline RL utilizing Q-functions can be further enhanced if it effectively handles OOD data generated by diffusion. The paper suggests improving IQL to train a value function using OOD but constrained actions, by instead using a skill prior learned by diffusion. Furthermore, the paper suggests adaptive re-evaluation, which re-plans the trajectory if the future value function becomes worse than the current value function. DIAR outperforms prior approaches in long-horizon, sparse-reward environments.

**Strengths:**

S1. DIAR consistantly outperforms prior works
* Table 1 shows that DIAR outperforms prior offline RL & diffusion-based offline RL works in 7 out of 9 tasks.

S2. Simplicity of proposed Adaptive Re-evaluation
* Adaptive Re-evaluation simply compares the future value function and current value function while decision making, preventing the policy from heading to worse states.
* With the assumption of goal-conditioned RL, the method is valid.

**Weaknesses:**

W1. Lack of novelty compared to LDCQ
* It seems that DIAR is an incremental improvement of LDCQ, which changed the base offline algorithm from BCQ to IQL. Specifically, DIAR uses the same procedure of LDCQ for getting the latent priors, using $\beta$-VAE for latent representation and getting the latent priors via diffusion.
* Please correct me if my understanding is incorrect in Q1.

W2. Restricted to goal-conditioned tasks
* Due to the sparse-reward assumption for the AR (adaptive re-evaluation), the application is limited to goal-conditioned tasks.
* Since DIAR is limited to goal-conditioned tasks, comparison with offline goal-conditioned RL algorithms will be insightful. Quick way to do improve this point will be adding the result of HIQL [1], which already have experimental results for AntMaze and Kitchen environments. If authors have enough time and resources, comparison with other offline GCRL methods such as GoFar [2], SMORe [3] will be informative.
* Additionally, it will be exciting if there is a way to generalize AR process to dense reward tasks, making the whole algorithm generally applicable.

W3. Effectiveness of re-evaluation
* According to Table 2, the performance improves in 5 tasks, and decreases in 4 tasks.
* Analyzing the failure cases of re-evaluation will be informative to further understand this behavior.

[1] Seohong Park, et al. HIQL: Offline Goal-Conditioned RL with Latent States as Actions, NeurIPS 2023

[2] Yecheng Jason Ma, et al. Offline Goal-Conditioned Reinforcement Learning via f-Advantage Regression, NeurIPS 2022

[3] Harshit Sikchi, et al. SMORE: Score Models for Offline Goal-Conditioned Reinforcement Learning, ICLR 2024

**Questions:**

Q1. Lack of novelty compared to LDCQ
* DAIR seems to be IQL version of LDCQ + Adaptive re-evaluation. Is this understanding correct?
* Would you mind to highlight the differences between LDCQ and DIAR?

Q2. Comparison with goal-conditioned RL method
* Can you compare DAIR with offline goal-conditioned RL methods (e.g. HIQL, GoFar, SMORe)?
* Comparison with goal-conditioned RL will be informative for those willing to apply DIAR for goal-conditioned tasks.

Q3. Why adaptive re-evaluation sometimes degrades the performance?
* The performance decreases in 4 out of 9 tasks when AR is applied.
* Examples for the failure cases of adaptive re-evaluation (e.g. Maze2D environments) and analysis for those will be informative to further understand the results.

Q4. Effectiveness of AR
* Can you apply AR for other methods (e.g. LDCQ, IQL) and see the improvements? One can apply AR for Q function to increase, if there is no value function in the method. Please do not hesitate to note the challenges if you have for applying AR to other methods.
* If you have any, can you share the idea of generalizing AR process to dense reward tasks? It will be exciting if there is a way to generalize AR process, making the whole algorithm generally applicable.

Q5. Loose bound of AR
* While deriving the formula of AR, it seems that the tight bound for $V(s_{t+H})$ is $\gamma^{-H} V(s_t)$.
* Can you share your thoughts on using a tighter bound $V(s_{t+H}) \geq \gamma^{-H} V(s_t)$ instead of $V(s_{t+H}) \geq V(s_t)$?

---

> ### Author Response · Authors · 2024-11-26
> **Response to Reviewer y9BV**
>
> **Q1.** DIAR is different from simply adding value function learning to LDCQ. Our goal was to make Q-function learning more precise, so we introduced a value function to ensure balanced training of the Q-function. By incorporating constraints from the value function during training, we can predict Q-values with greater accuracy. Additionally, the diffusion model, as a generative model, can produce a wide variety of meaningful latent vectors. We aimed to leverage this capability to enhance both Q-function and value-function learning. In the Bellman equation for Q-function learning in LDCQ, $Q(s_t,z) ← (r_{t:t+H}+\gamma^H Q(s_{t+H}, argmax(Q(s_{t+H},z_i)))$, the reward term $r_{t:{t+H}}$ is undefined if $z$ is not sampled from the offline dataset. However, by introducing the value function, we can indirectly evaluate the value of latent vectors generated by the diffusion model. As a result, DIAR uses not only the offline dataset but also latent vectors sampled from the diffusion model in the training process for both Q-functions and value functions. This approach led to significant performance improvements on datasets such as Maze2D, AntMaze, and Kitchen.
>
> **Q2.** Our goal was to address offline reinforcement learning problems using a diffusion model. To this end, we compared DIAR with algorithms like Diffuser and DD, as well as skill-based algorithms. While the tasks in our experiments focused on goal based tasks, we did not explicitly involve providing the goal as input or solving tasks where the objective is to directly reach a specified destination. However, since DIAR can also function as a goal-conditioned RL algorithm, it would be possible to include the goal as part of the input and conduct additional experiments for comparison under goal-conditioned settings. Evaluating the ability to find an optimal path from the current state to a specified goal is as important as the experiments we have conducted so far and could offer further insights into DIAR’s capabilities.
>
> **Q3.** The value function is used to evaluate whether the current decision is appropriate, and if it is not, a new decision is generated. However, it is challenging to train the value function perfectly. If the value function fails to accurately assess the value of the current state, the new decision generated may not be more optimal. Additionally, since we do not have access to unlimited data, this limitation can pose challenges for the DIAR model. Currently, AR is designed based on a few assumptions, but we plan to refine it into a more general algorithm in the future, making it applicable to a wider range of tasks and domains.
>
> **Q4.** The purpose of introducing AR was to address the possibility of finding a more optimal choice when selecting a long action sequence all at once. Therefore, if there is a method to evaluate the current decision, AR can be applied even without relying on the value function. Additionally, AR can be used in environments with dense rewards. However, to enable more general applicability, the algorithm needs to be further expanded and refined.
>
> **Q5.** As the reviewer mentioned, a tighter bound for the value is indeed $\gamma^{-H}V(s_t)$. In Section 4.3, the bound is calculated using $\gamma^{-H}V(s_t)$. However, when applying AR, we did not explicitly use $\gamma$. Instead, we evaluated the latent vectors' values based on the value function $V(s_t)$. To account for the possibility of a noisy value function, we incorporated a slight margin, allowing AR to operate effectively when there is a significant difference in value. When the value differences are small, the algorithm tends to follow the original decision more closely, maintaining a balance between exploration and adhering to the initial choice.

---

> ### Comment · Reviewer_y9BV · 2024-11-27
>
> **Q1.**
>
> Thank you for clarifying the distinction between DIAR and LDCQ. I now have a clearer understanding of the motivation behind incorporating the value function and the differences between the two approaches.
>
> However, I still perceive DIAR as an "IQL version of LDCQ" combined with "adaptive revaluation." While the authors argue that DIAR is distinct from IQL + LDCQ, the critic updates closely mirror those of IQL, with the primary difference being the skill-based approach. Additionally, the diffusion latent extraction appears to be derived from LDCQ. Specifically, lines 251–264 seem to follow the IQL approach with $z\_t$ replacing $a\_t$ (I would strongly recommend citing IQL in relation to the expectile loss here).
>
> While combining existing methods is a valid and valuable contribution, it typically requires strong and broad empirical support to compensate for limited novelty.
>
> &nbsp;
>
> **Q2.**
>
> I appreciate the clarification that DIAR is not a goal-conditioned RL algorithm. However, its applicability appears largely restricted to goal-conditioned tasks. Without demonstrating its performance in more diverse environments, comparisons with general-purpose algorithms may be unfair, as DIAR seems particularly tailored to goal-conditioned tasks.
>
> &nbsp;
>
> **Q3, Q4, Q5.**
>
> Thank you for addressing the practical considerations surrounding adaptive revaluation (AR).
> I agree that refining adaptive revaluation into a more generalized framework is a promising direction for future work.
> However, applicability of AR still remains limited, and effectiveness of AR is not strongly demonstrated in the current version.
>
> &nbsp;
>
> Despite the clarifications, I still have two remaining concerns:
> * **Novelty**: The critic updates largely follow IQL, and the diffusion latent extraction process is derived from LDCQ.
> * **Effectiveness of Adaptive Revaluation**: The adaptive revaluation, which is the key novelty of the paper, demonstrates limited applicability, and its empirical effectiveness remains unclear.
>
> Due to these unresolved concerns, I will maintain my current score.

---

### Official Review · Reviewer_f1C1 · 2024-10-31

**Soundness:** 2
**Presentation:** 2
**Contribution:** 2
**Rating:** 5
**Confidence:** 4

**Summary:**

The paper presents a novel offline reinforcement learning (RL) framework that leverages diffusion models to address challenges such as out-of-distribution samples and long-horizon decision-making. By introducing an Adaptive Revaluation mechanism, the DIAR framework dynamically adjusts decision lengths based on current and future state values, enhancing long-term decision accuracy. The Q-value overestimation  is mitigated through the generation of diverse latent trajectories. Empirical results on tasks like Maze2D, AntMaze, and Kitchen demonstrate that DIAR consistently outperforms state-of-the-art algorithms, underscoring its potential for real-world applications in robotics and autonomous systems.

**Strengths:**

1. The introduction of the Adaptive Revaluation mechanism is novel. It dynamically modifies decision lengths based on comparative state values, enhancing decision-making flexibility.
2. This approach is systematically evaluated, with performance improvements validated by extensive empirical results, as shown by the statistically significant outperformance of DIAR over comparable models.
3. The structure of the paper is well-organized and logical, breaking down the methodology, implementation, and adaptive mechanisms in detail.

**Weaknesses:**

1. The experiments are primarily conducted on three well-known offline RL environments (Maze2D, AntMaze, and Kitchen) of total 9 datasets. While these provide a foundation, they are limited in diversity, which could restrict understanding of DIAR’s generalizability. Including more datasets makes the environmental results more persuasive.
2. Adaptive revaluation is proposed as a mechanism to improve decision flexibility, yet its theoretical grounding is somewhat limited. For instance, the model could benefit from a more rigorous analysis of how adaptive revaluation specifically affects trajectory optimization, particularly in long-horizon scenarios where trajectory value predictions might become noisy or overly optimistic.
3. The mixed training of Q and V, as well as the use of weighted training techniques, has not been evaluated through an ablation study, making it unclear what their contributions are.
4. There is no sensitivity analysis on the hyper-parameters, such as $\tau$ and $\beta$.

**Questions:**

1. The article states that the mixed training of Q and V reduces overestimation issues by utilizing latent states generated by the diffusion model. However, I question the effectiveness of this approach in mitigating overestimation, as the latent states from the diffusion model are also not part of the original data. Additionally, there is no pessimistic approach incorporated into the value function during this process. How can this be justified?

---

> ### Author Response · Authors · 2024-11-26
> **Response to Reviewer f1C1**
>
> **Weakness 1.**  In this study, we focused on long-horizon sparse reward problems, such as those in Maze2D, AntMaze, and Kitchen. Accordingly, the methods proposed in DIAR were designed to perform well in these environments, achieving significant performance improvements. However, we aim to expand our approach in the future to work effectively in other environments, such as  half-cheetah, walker2d and hopper, by developing more generalizable ideas. We are always open to new ideas or discussions about extending the DIAR algorithm to a broader range of tasks.
>
> **Weakness 2.** It is challenging to train a value function without any errors. A noisy value function struggles to accurately assess the value of a state, which can hinder the ability to make optimal decisions. In fact, experiments analyzing the impact of AR show that it improves performance in some cases but causes slight decreases in others. Additionally, since we do not have access to unlimited data, this limitation can make it difficult for the DIAR model to perform well in certain scenarios. Currently, AR is designed based on several assumptions, but we plan to refine it into a more general algorithm in the future, enabling its application across a wider range of domains.
>
> **Weakness 3 & Q1.** DIAR is not simply an extension of LDCQ with a value function added. Our goal was to make Q-function learning more precise, and to achieve this, we introduced a value function to ensure balanced training of the Q-function. By incorporating constraints derived from the value function, we can predict Q-values with greater accuracy. Additionally, the diffusion model, as a generative model, can produce a diverse range of meaningful latent vectors. We sought to leverage this capability for both Q-function and value-function learning. In the Bellman equation used for Q-function learning in LDCQ, $Q(s_t,z) ← (r_{t:t+H}+\gamma^H Q(s_{t+H}, argmax(Q(s_{t+H},z_i)))$, the reward term $r_{t:{t+H}}$ is undefined if $z$ is not sampled from the offline dataset. However, by introducing the value function, we can indirectly evaluate the value of latent vectors sampled from the diffusion model. As a result, DIAR incorporates not only the offline dataset but also latent vectors generated by the diffusion model in the training process for Q-functions and value functions. Using this approach, we demonstrated significant performance improvements on datasets such as Maze2D, AntMaze, and Kitchen.
>
> **Weakness 4.** According to the LDCQ paper, a fixed value for $\beta$ is used, highlighting its advantage of being less sensitive to various hyperparameters. In our current experiments, we also use a fixed value of 0.9 for $\tau$. However, as the reviewer suggested, analyzing the sensitivity of parameters like $\tau$ could be a valuable direction for further investigation.

---

> > ### Comment · Reviewer_f1C1 · 2024-11-30
> >
> > Thank you for the authors' response. I appreciate the clarification that DIAR is different from LDCQ. However, the introduction of the value function and the use of the diffusion model to generate latent variables are techniques that have been attempted in previous works. The article combines these techniques, and it seems that the novelty is still insufficient to support the claim. Therefore, I maintain my score.

---

### Official Review · Reviewer_VqTk · 2024-11-01

**Soundness:** 2
**Presentation:** 2
**Contribution:** 2
**Rating:** 3
**Confidence:** 5

**Summary:**

This paper presents a diffusion-guided offline reinforcement learning method, DIAR, designed to address challenges posed by out-of-distribution samples and long-horizon planning. Specifically, DIAR first trains a VAE to extract trajectory representations, which are then used as generation targets for training a corresponding diffusion model. DIAR subsequently leverages the representations generated by the diffusion model to support the learning of the value function and policy network. Finally, the authors validate the effectiveness of DIAR through experiments on sparse reward tasks.

**Strengths:**

1. Offline RL problem in sparse reward tasks is significant, and previous methods have struggled to effectively address it.
2. Compared to the baselines, DIAR demonstrates superior overall performance on sparse reward tasks.

**Weaknesses:**

1. The writing in this paper could be improved, as the logic is not always coherent, and some sentences are difficult to understand. For example, “However, offline RL relies on the given dataset, so the learned policy may be inefficient or misdirected if the data is poor quality or biased.” in lines 44-46.
2. The novelty of the method is limited, as both learning a latent diffusion model [1] and implicit Q-learning [2] are existing approaches.
3. The use of the diffusion model lacks clear motivation. The authors need to further explain why it is necessary to input the latent representation $z$ during the IQL stage.
4. The adaptive revaluation approach is not entirely reasonable, as expert policies in many tasks do not satisfy $V(s_{t+1}) > V(s_t)$. For example, this is often the case in finite-horizon tasks with positive reward functions. Consequently, the theoretical analysis in the paper relies on overly strong assumptions that do not generalize well to other tasks.

[1] Reasoning with Latent Diffusion in Offline Reinforcement Learning. ICLR, 2024.

[2] Offline Reinforcement Learning with Implicit Q-Learning. ICLR, 2022

**Questions:**

See weaknesses.

---

> ### Author Response · Authors · 2024-11-26
> **Response to Reviewer VqTk**
>
> **Weakness 1.** We reviewed and revised sentences from the paper that were difficult to understand. The revised content is highlighted in blue for clarity.
>
> **Weakness 2.** As the reviewer mentioned, LDM and IQL are existing methods in their original forms. In our model, DIAR, LDM serves two purposes: (1) generating candidate latent vectors for decision-making and (2) aiding in Q-function training. However, our model is not simply a combination of IQL and LDM. We aimed to make Q-function learning more precise by introducing a value function, which ensures balanced training of the Q-function. By training the Q-function with constraints derived from the value function, we can achieve more accurate Q-value predictions. Additionally, the diffusion model, as a generative model, can produce a diverse range of meaningful latent vectors. We sought to leverage this capability for both Q-function and value-function training. In the Bellman equation used for Q-function learning in LDCQ, $Q(s_t,z) ← (r_{t:t+H}+\gamma^H Q(s_{t+H}, argmax(Q(s_{t+H},z_i)))$, the reward term $r_{t:{t+H}}$ is unknown if $z$ is not sampled from the offline dataset. However, by incorporating the value function, we can indirectly evaluate the value of latent vectors sampled from the diffusion model. Thus, DIAR utilizes both the offline dataset and latent vectors generated by the diffusion model during Q-function and value-function training. With this approach, we demonstrated performance improvements on datasets such as Maze2D, AntMaze, and Kitchen.
>
> **Weakness 3.** The need to incorporate a latent diffusion model (LDM) is also highlighted in the LDCQ paper for several reasons:
> - Flexible decoder design: Since the latent diffusion model operates in the latent space, even discrete action spaces can be effectively represented. This allows for more flexibility in designing the decoder.
> - Temporal abstraction: With the help of LDM, a powerful generative model, it is possible to create a temporally abstract and information-dense latent space.
> - Faster training and inference: Generating latent vectors is much more efficient than directly generating action-state sequences, as done in traditional methods, leading to faster training and inference processes.
>
> **Weakness 4.** In Maze2D and AntMaze, rewards are given only when the agent reaches the goal, and all other states yield a reward of 0. Under the assumption of an expert policy, the value increases steadily as the agent gets closer to the goal. However, if additional rewards are provided for states outside the goal region, this condition may no longer hold, potentially leading to performance degradation when using AR. In the case of Kitchen, there are several intermediate sub-goals, and rewards are given when these sub-goals are achieved. In such scenarios, it can be challenging to make optimal decisions using AR based solely on the value function. We would greatly appreciate any ideas, suggestions, or feedback on AR, as they could significantly help us improve our algorithm.

---

> > ### Comment · Reviewer_VqTk · 2024-11-28
> >
> > Thank you for your detailed response. I appreciate the improvement for IQL by incorporating the diffusion-generated latent vector, and I believe this is a reasonable approach. However, I still think there are several issues with the work:
> >
> > 1. The presentation still needs improvement. I suggest that the authors highlight the best result for each task in the table and calculate the average performance of each method across. Additionally, I recommend a more detailed summary and analysis of the experimental results. For example, by comparing the results of IQL and DIAR, we can observe the contribution of the latent vector in learning more accurate Q-values.
> >
> > 2. My main concern lies in the limitations of adaptive revaluation. Adaptive revaluation can only be applied to tasks where the value function monotonically increases with the timestep, such as the AntMaze and Maze2d tasks listed by the authors. However, the remaining locomotion tasks in D4RL do not meet this condition.
> >
> > Therefore, I maintain my score.

---

### Official Review · Reviewer_XDtn · 2024-11-04

**Soundness:** 2
**Presentation:** 1
**Contribution:** 1
**Rating:** 3
**Confidence:** 4

**Summary:**

The paper proposes an offline RL algorithm utilizing latent diffusion skill models for temporal abstraction, and Q learning with these skills. During policy rollouts, the learnt value function and temporally abstract world model are used to evaluate whether the currently used skill is optimal. If not, a new skill latent is selected. The method is demonstrated on D4RL tasks.

**Strengths:**

1. The method has good performance on D4RL compared to baselines.

**Weaknesses:**

1. The paper is messily written in my opinion, and it is difficult to parse what the major contribution of the paper is. This seems like primarily an engineering paper, but this is not clearly communicated.
2. The method is a combination of existing offline RL algorithms (primarily LDCQ and IQL), but there is no proper reason given for this particular configuration of components. The only novel addition seems to be the use of the value function for deciding when to stop executing a skill, but this is a simple iterative improvement.
3. Novelty is not strictly necessary, but the additions made here are not well justified at all with no coherent story surrounding it.
4. This is not a direct criticism of the paper, but D4RL has been quite over-optimized in the offline RL community now, small engineering improvements to boost the score in this benchmark does not give any signal to the true value of the method.
5. More general writing criticism, a lot of the paper repeats itself and feels like padding more than informative content. For example, section 4.3 “Theoretical Analysis of DIAR” is very elementary and adds no value.

**Questions:**

1. What is the primary contribution of the paper? Do the authors pitch the paper as a novel offline RL algorithm?
2. Since the authors only evaluate on D4RL, why is there no evaluation of the locomotion tasks (half-cheetah, walker2d, hopper)?

---

> ### Author Response · Authors · 2024-11-26
> **Response to Reviewer XDtn**
>
> **Weakness 1,5.** We have reviewed the sentences that were difficult to understand in the paper and revised them to make the content easier to understand. The revised content is marked in blue text.
>
> **Weakness 2,3 & Q1.** Our approach is different from simply introducing value function learning into LDCQ. We aimed to make Q-function learning more precise by incorporating a value function, which helps ensure balanced training of the Q-function. By providing constraints from the value function during training, the Q-function can predict Q-values with greater accuracy. Additionally, the diffusion model, as a generative model, is capable of producing diverse and meaningful latent vectors. We wanted to leverage the diffusion model's ability to generate a variety of samples for both Q-function and value function learning. In the Bellman equation for Q-function learning in LDCQ, $Q(s_t,z) ← (r_{t:t+H}+\gamma^H Q(s_{t+H}, argmax(Q(s_{t+H},z_i)))$, the reward term $r_{t:{t+H}}$ cannot be determined if $z$ is not sampled from the offline dataset. However, by introducing the value function, we can indirectly evaluate the value of latent vectors sampled from the diffusion model. As a result, DIAR uses not only the offline dataset but also latent vectors sampled from the diffusion model in the training process for Q-functions and value functions. With this approach, we demonstrated performance improvements on datasets such as Maze2D, AntMaze, and Kitchen.
>
> **Weakness 4.** We agree with the reviewer’s observation that many optimal algorithms have been proposed for D4RL tasks. However, as shown in our comparative analysis (e.g., Maze2D), there is still significant room for performance improvement. Nevertheless, we believe it is important to further validate the algorithm's performance across various environments. In the future, we plan to evaluate our model on a wider range of datasets to better assess its effectiveness.
>
> **Q2.** In this study, we focused on long-horizon sparse reward problems, such as those in Maze2D, AntMaze, and Kitchen. Accordingly, the methods proposed in DIAR were specifically designed to perform well in these environments, and they demonstrated significant performance improvements. However, we aim to expand our approach in the future to incorporate more general ideas, enabling strong performance in other environments, such as half-cheetah, walker2d and hopper.

---

> > ### Comment · Reviewer_XDtn · 2024-12-01
> >
> > I thanks the authors for their response, however I still think the addition of the Value function makes this a combination of IQL and LDCQ, which is too small a change in my opinion. I will keep my score.

---

### Author Response · Authors · 2024-11-26
**General Response**

Dear Reviewers and Meta Reviewer,

Thank you for taking the time to review our submission, "DIAR: Diffusion-model-guided Implicit Q-learning with Adaptive Revaluation." We deeply appreciate your thoughtful feedback and constructive suggestions, which have greatly contributed to improving the quality of our work. Below, we provide a detailed response to each of your comments.

- We have provided a more detailed explanation of our concept, DIAR, and clarified the distinctions between our approach and existing methods.
- This study focuses on addressing the challenges of long-horizon tasks with sparse rewards, and our approach was specifically designed to tackle these issues. Through experiments, we demonstrated that our method performs well in environments such as Maze2D, AntMaze, and Kitchen.
- We tested Adaptive Revaluation (AR) under scenarios that required certain assumptions, and we are considering extending it to a broader range of applications in the future.
- We revised complex sentences in the paper, improving their clarity to make them easier for readers to understand.

Should you have any further questions or suggestions, please put your comments on OpenReview. We will address all the raised concerns according to the reviewing policy.

---

### Meta-Review · Area_Chair_JVEV · 2024-12-15

**Metareview:**

Authors present a model-based offline RL method using diffusion models that outputs sequence-level distributions to handle long horizons and deal with compounding error issues that occur with 1-step models. Their method also addresses out-of-distribution samples with a learned value function.

Strengths: DIAR has strong performance on D4RL and the adaptive reevaluation component is novel.
Weaknesses: The paper clarity needs improvement and contribution appears weak -- it combines components from existing works and evaluation is performed on a narrow set of environments.

Overall, the paper still seems quite immature -- the writing needs improvement and most of the contributions come from existing works.  The adaptive revaluation, which is the key novelty of the paper, demonstrates limited applicability, and its empirical effectiveness remains unclear. For these reasons, I vote to reject.

**Additional Comments On Reviewer Discussion:**

Authors were unable to satisfactorily address the main concerns of reviewers in the rebuttal phase, leading reviewers to maintain their scores.

---

### Decision · Program_Chairs · 2025-01-22

Reject